# Association of Food Intake with Sleep Durations in Adolescents from a Capital City in Northeastern Brazil

**DOI:** 10.3390/nu14235180

**Published:** 2022-12-05

**Authors:** Emanuellen Coelho da Silva, Juliana Ramos Carneiro, Poliana Cristina de Almeida Fonseca Viola, Susana Cararo Confortin, Antônio Augusto Moura da Silva

**Affiliations:** 1Department of Public Health, School of Nutrition, Federal University of Maranhão, São Luís 65020-905, MA, Brazil; 2Department of Public Health, School of Medicine, Federal University of Maranhão, São Luís 65020-905, MA, Brazil; 3Nutrition Department, Nutrition Teacher at the Health Sciences Center, Federal University of Piauí, Teresina 64049-550, MA, Brazil; 4Postgraduation Program in Collective Health, Department of Public Health, Federal University of Maranhão, São Luís 65020-905, MA, Brazil

**Keywords:** sleep duration, eating habits, adolescence, ultra-processed foods, quality of life

## Abstract

(1) Background: During adolescence, there are significant changes in food consumption, such as reducing the consumption of *in natura* or minimally processed foods and increasing the consumption of ultra-processed foods. Thus, eating habits can influence sleep duration and, consequently, affect the quality of life of young people. This study thus aims to estimate the association of consumption of *in natura* or minimally processed, processed, and ultra-processed foods with sleep durations in adolescents. (2) Methods: This is a cross-sectional study including 964 adolescents (18 to 19 years old) from the 1997 to 1998 birth cohort in São Luís, Maranhão. Food consumption was assessed using the food frequency questionnaire (FFQ) and stratified based on the NOVA classification. Sleep duration was verified using accelerometry in hours. The analysis of the association between the consumption of *in natura* or minimally processed, processedand ultra-processed foods with sleep durations in adolescents used crude and adjusted linear regression (by gender, age, skin color, education, economic class, work, consumption of alcohol, smoking, screen time, physical activity, use of illicit drugs, anxiety, depressive symptoms, and lean and fat mass). A directed acyclic graph (DAG) was used to determine the minimum set of adjustment factors. (3) Results: Of the 964 individuals evaluated, 52.0% were female. The mean sleep duration was 6 h (± 0.95). In the crude and adjusted analyses, no association was observed between food consumption according to the degree of processing and adolescent sleep durations. (4) Conclusion: There was no association between the consumption of *in natura* or minimally processed, processed, and ultra-processed foods with sleep durations.

## 1. Introduction

Sleep duration plays a crucial role in different life cycles. Sleep is essential in adolescence for physical growth, development, and socio-emotional wellbeing [1]. In the US, the National Sleep Foundation (NSF) recommends 8 to 10 h of sleep at night for adolescents. Sleep exerts several physiological functions via memory consolidation, thermoregulation, and the restoration of energy metabolism [2], which are vital phenomena during which mechanisms and organic systems are “turned off” or attenuated, aiming at the prevention of exhaustion [3]. In this regard, many studies analyze sleep durations using more conventional methods, such as a self-reported sleep questionnaire [4].

The accumulation of insufficient sleep durations, throughout adolescence, may cause behavioral and physiological alterations that reflect risk factors for diseases [5]. Thus, the habit of sleeping few hours during the night influences negatively all systems of the human body, causing cardiovascular diseases [6], a reduction in cognitive functions [7], brain deficits, and neuropsychiatric disorders [8], among other damages.

Some factors may contribute to an increase in the deleterious effects of sleep duration [5], such as demographic, socioeconomic [3,9], and behavioral [3] characteristics and food choices [10]. There is evidence that the high consumption of ultra-processed foods [11] may lead to reduced sleep durations in adolescents. In view of the above, some longitudinal studies in the literature address the relationship between food intake and sleep duration. Sousa et al. [4] showed that the higher consumption of ultra-processed foods was a risk factor for poor sleep quality. Thus, individuals who consume caloric foods [10] are more likely to sleep less. In this sense, studies that investigated this association attribute such findings to the total energy intake, relating macro- and micronutrients [4,11]. Another hypothesis is that increased caloric intakes may cause neurohormonal changes that influence the reduced hours of sleep [12,13,14]. From this perspective, the increased consumption of high-calorie density foods is related to increased inflammatory responses [15,16]. As a consequence, there may be a deterioration in the quality of health, as it contributes to overweight and obesity, in addition to causing hormonal, metabolic, and physiological alterations that lead to worsening sleep quality [17,18,19]. Thus, insufficient sleep durations are possibly associated with a high intake of refined carbohydrates [20], total fat, monounsaturated fat, trans fat, saturated fat, and polyunsaturated fat [21]. In addition, the high consumption of ultra-processed foods is linked to anxiety-induced sleep disorders [22] since the intake of these foods is related to an increase in inflammatory markers, which can lead to insomnia [23]. On the other hand, some foods, such as those rich in tryptophan and tyrosine, are involved in the production of serotonin and melatonin and can help maintain adequate sleep [10].

No published studies to date evaluate the association between food intake according to the degree of processing with sleep durations assessed by accelerometry. Given this, it is known that the accelerometer is an objective measure widely used in epidemiological studies, unlike the questionnaire, which is a subjective measure that has inherent limitations relative to self-reported methods [24]. However, actigraphy is the most suitable tool for sleep–wake detection, it is useful in diagnosing sleep-related disorders such as circadian cycle disorders, and it is the most reliable tool for analyzing sleep durations for longitudinal studies compared to the application of sleep questionnaires [25]. Thus, it is necessary to make use of this tool to investigate the relationship between food intake and sleep durations, as such information may be useful in providing health interventions for these individuals, enabling prevention strategies aimed at proper diet and sleep in adolescence and later in life. Given this, the present study aims to estimate the association of consumption of *in natura* or minimally processed, processed, and ultra-processed foods with sleep durations in adolescents from a Brazilian birth cohort.

## 2. Methods

### 2.1. Study Design and Population

This cross-sectional study was conducted with adolescents aged 18 to 19 years who are a part of the third phase of the 1997/1998 São Luís Birth Cohort, titled “Determinants throughout the life cycle of obesity, precursors of chronic diseases, human capital and mental health: a contribution of the São Luís birth cohorts to SUS”. The São Luís Birth Cohort baseline included a systematic sample, with 1/7 of births in 10 maternity hospitals of São Luís residents selected, with a final sample of 2493 live births. In 2005/2006, the first follow-up was conducted, in which 673 children aged 7 to 9 years participated. In 2016/2017, the second follow-up was conducted with adolescents aged 18 to 19 years, with the attempt to evaluate the entire original cohort. A retrospective component was included in this follow-up study (with the application of a key part of the perinatal questionnaire to mothers of adolescents), which included 1828 adolescents born in São Luís in 1997 who had not participated in the original cohort. The new members were selected by random sampling from the Live Births Information System restricted to children born in 1997. Adolescents identified in schools and universities and via as social network were also included. Thus, 2515 adolescents participated in this follow-up study [26].

However, for the analytical sample of this study, data from 964 adolescents were used, considering that 1538 individuals made use of the accelerometer; however, 179 were excluded for not having data in the software and 11 of these were excluded for calibration errors (calib_err > 0.02), 7 were excluded for having a non-zero clipping score, 3 were excluded for not having a complete 24 h cycle, 14 were excluded for inadequate plots, and 4 more adolescents were excluded for not having correct or complete identification. The final accelerometer sample was 1324, but data from adolescents with less than two hours of sleep were excluded and all missing adjustment variables were removed (Figure 1).

### 2.2. Dependent Variable

The study’s dependent variable, sleep duration, was analyzed using an accelerometer (model GTX3+, Actigraph^®^; Pensacola, FL, USA) with a previous calibration. This tool is valid and reliable, with a clear ability to provide data regarding measurements that are useful during the sleep–wake cycle by detecting limb movements for 24 h. Although many of the subjective instruments used are validated questionnaires, for example, Karolinska Sleep Diary, Pittsburgh Sleep Diary, and Sleep Timing Questionnaire, they present a moderate or low correlation when compared to the gold standard instruments for sleep durations, such as polysomnography or actigraphy [27,28]. Although polysomnography is considered the gold standard method for investigating sleep-related variables, including duration, the use of accelerometers/actigraphy gained strength for sleep assessments in epidemiological research [27,28]. The use of polysomnography, the gold standard method, and the use of the accelerometer provide greater logistical ease for research, since they are less invasive methods that do not require the displacement of the subject to the laboratory; moreover, these methods make it possible to measure sleep over a longer period of time [29].

The adolescents participating in the survey were asked to use the accelerometer for up to seven consecutive days without differentiating the weekend. In addition, they were instructed to use the instrument for twenty-four hours a day on the non-dominant wrist and were instructed to remove the device when in the bath and when carrying out activities in a water environment.

Regarding this procedure, data were collected in 5 s epochs with a frequency of 60 Hz. Raw data extraction from the accelerometer was performed using ActiLife software version 6.12. Subsequently, such data were processed using the R statistical package (GGIR package version 1.11-0), in which non-human movements were filtered, usage time was validated, and a recalibration process was carried out for each datum. Furthermore, the process constructed data quality graphs for each participant in the analysis. While data from 1363 participants were submitted, some of these were excluded: those who had some errors or defects in the data (*n* = 35) due to a calibration error greater than 0.02, patterns not compatible with human movements, less than an entire 24 h cycle, and those with quality problems after a visual inspection of the graphs (*n* = 84) and incomplete data, which may refer to less than 4 days of use.

The reliability test for the accelerometer’s minimum days and sleep parameters used the Spearman–Brown formula [30]. Thus, a minimum of 4 nights was defined, with 0.52 as the best reliable result concerning the loss of many observations. Moreover, an algorithm was used to identify sleep parameters [31] and automatic detection [32]. In this regard, sleep episodes were defined according to periods of inactivity sustained in the sleep period window since the algorithm detected the sleep bases with respect to changes in the angle compared to the horizontal plane (*z*-axis) [32]. The analyses used a time window of 5 min and a change in wrist angle of 3 degrees to detect sleep parameters. Thus, the present study will use the variable total sleep duration in hours, which results from the total minutes classified by the algorithm as sleep.

### 2.3. Independent Variables

Food consumption was assessed using the food frequency questionnaire (FFQ) developed by Schneider et al. [33] and adapted and validated for adolescents in São Luís [34]. The FFQ is composed of 106 foods or food groups, with 8 consumption frequency options (never or < 1 time/month; 1 to 3 times/month; 1 time/week; 2 to 4 times/week; 5 to 6 times/week; 1 time/day; 2 to 4 times/day; ≥5 times/day) and the reference average portion size of foods. For each food, adolescents were asked about consumption frequency and the portion usually consumed (whether the portion is smaller, equal, or larger than the reference) [33].

The QFA was applied by nutritionists and nutrition students, with the aid of a photo album to help estimate the portions consumed. The interviewers were trained to apply the QFA by nutritionists and professors of the nutrition course at the Federal University of Maranhão (UFMA). During the training sessions, the simulations of QFA applications were performed on the adolescents of the same age group in the study’s population. This protocol is widely used in epidemiological studies that investigate the relationship between diet and chronic noncommunicable diseases, and it is considered the most practical and informative tool for assessing dietary profiles. In this regard, each food’s consumption frequency was converted into daily frequency and associated with the size of the consumed portion to calculate the energy value of each food item and the total daily energy consumption of the diet. Subsequently, the nutritional value of the foods consumed was obtained based on the Brazilian Food Composition Table [35] and the Nutritional Composition Table of Foods Consumed in Brazil [36] added information from food labels and regional recipes.

According to the NOVA classification, the FFQ food items were classified into three groups according to the Dietary Guidelines for the Brazilian Population. The three food groups are (1) *in natura* or minimally processed foods; (2) processed foods; and (3) ultra-processed foods. Above all, culinary preparations based on one or more in natura or minimally processed foods were classified in the first group [37]. Finally, the percentage contribution of each food group to the total caloric value (TCV) of the diet was calculated, thus obtaining the percentage of calories from each food group, which were categorized into distribution tertiles.

### 2.4. Adjustment Variables

The study’s complementary variables were related to socioeconomic, behavioral, and health characteristics. The included variables were the following: gender (female and male), age (18 and 19 years old), and the self-reported skin color, which was assessed by the Survey on Ethnic-Racial Characteristics of the Population-PCERP/IBGE 2008. This investigation starts the block on color or race and gradually approaches the interviewee with the theme of ethnic–racial self-identification and presenting him/her with the elements that are recognized as constitutive of these processes of identity elaboration via the self-reported skin color (white, black, and brown). One of the key issues in the study of ethnic–racial identification of the population refers to the multidimensionality of this phenomenon [38]. Years of schooling (0 to 8 years, 9 to 11 years, and 12 years or older), socioeconomic classification according to the criteria of the Brazilian Association of Research Companies—ABEP (A, B, C, and D-E) [39]—and family income in minimum wages (≤3 minimum wages, 3.1 to 6 minimum wages, and >6 minimum wages) (minimum wage was equivalent to BRL 880.00 in 2016, BRL 937.00 in 2017, and BRL 954.00 in 2018) were recorded.

As for lifestyle, physical activity was assessed using the Self-Administered Physical Activity Checklist (SAPAC) [40]. Those that were insufficiently active had less than 300 min of physical activity per week, and physically active individuals had 300 min or more per week. Screen time was measured as less than or equal to 5 h a day and greater than 5 h a day of exposure to electronic devices such as cell phones, tablets, computers, video games, and television during weekdays. Weekends were disregarded in the evaluation [1] because weekend use exceeds the usual exposure reported on weekdays, and this could affect the analysis on adolescents’ screen time. The current alcohol consumption (no or yes) was assessed by the Alcohol Use Disorders Identification Test (AUDIT) [41], current smoking (no or yes), and illicit substance use (never used, previously used, or currently using). For the classification of anthropometric status of adults, the World Health Organization’s [42] cut-off points were adopted; BMI <18.5 kg/m^2^ (underweight)/BMI >18.5 to 24.9 kg/m^2^ (eutrophy) and BMI ≥25 to 29.9 kg/m^2^ (overweight)/BMI > 30.0 kg/m^2^ (obesity). MINI (Mini International Neuropsychiatric Interview-Brazilian version 5.0.0-DSM IV) [43] was used to assess major depressive episodes or depression (yes/no) and anxiety (yes/no).

### 2.5. Data Analysis

Descriptive analyses were performed for all variables, estimating the means and standard deviations for sleep durations according to exposures and complementary variables. The mean values of the outcome concerning the complementary variables were compared using Student’s *t*-test, Mann–Whitney U test, one-way ANOVA, or Kruskal–Wallis test. Pearson’s correlation was performed to verify the relationship between food consumption and the outcome.

Regarding the covariates of the study, a directed acyclic graph (DAG) was constructed in the DAGitty^®^ version 3.0 program based on the literature review on the theme of the study, which demonstrated a relationship between food intake and the duration of sleep (Figure 2). The DAG is called an influence diagram, relevance diagram, or causal network because the arrowheads in the graph create a path between two variables. It is worth noting that the the main characteristics of these graphs are as follows: being acyclic, that is, not allowing circularities in its composition. This means that a variable cannot interfere with itself and is directed, where causality follows only one direction, at a given moment in time. By assuming the temporal perspective in relation to the notion of causality, one has it that time flows from the left to the right [44].

The DAG is a visual tool that encodes qualitative expert knowledge or assumptions about the causal structure of a problem and serves to represent a directed causal relation-ship between confounding variables identified via theoretical grounding, constructing a minimal set for the adjustment variables [45]. Simply put, the DAG is a visual tool that is able possible to encode knowledge (from both empirical research as well as theory) regarding the causal structure of a problem. Moreover, from the observation of the created design, it is possible to identify biases (of confounding and variable selection), thus helping in the definition of which variables should be included in the adjustment so as to not sabotage the identification/interpretation of the causal effect [46]. To investigate the association between the consumption of *in natura* or minimally processed, processed, and ultra-processed foods and sleep, the following minimal set of adjustment variables was selected: sex, age, skin color, education, economic class, work, alcohol consumption, smoking, screen time, physical activity, illicit drug use, anxiety, depressive symptoms, lean mass, and fat mass.

Linear regression (crude and adjusted) was used to analyze the association between the consumption of *in natura* or minimally processed, processed, and ultra-processed foods with sleep durations in adolescents, calculating the regression coefficient and respective 95% confidence interval (95% CI). Statistical analyses were performed using Stata software version 14.0.

## 3. Results

In the distribution of participants, according to use or not of an accelerometer, there were differences between age categories (*p* ≤ 0.001), years of schooling (*p* = 0.011), economic class (*p* = 0.001), drug use (*p* = 0.048), depression (*p* ≤ 0.001), and the consumption of processed foods (*p* = 0.024) (Table 1). Most participants who used the accelerometer were 18 years old, had 9 to 11 years of schooling, were from economic class C, and were not drug users (Table 1).

A total of 964 individuals were evaluated, of which 52.0% were female, 85.9% were 18 years old, 65.3% declared themselves to have brown skin color, 90.4% had 9 to 11 years of schooling, 50.0% belonged to economic class C, 33.8% had a family income of 1 minimum wage, and 84.3% did not work. Regarding lifestyle, 55.4% were insufficiently active, 61.7% used a screen more than 5 h a day, 59.9% consumed alcoholic beverages, 3.1% used tobacco, 23.6% used or use illicit drugs, 3.2% had anxiety, 11.0% had depressive symptoms, and 19.8% were overweight or obese (Table 2). The mean sleep duration was 6 (± 0.95) hours. The consumption of the average daily percentage caloric value for *in natura* or minimally processed foods was 34.72% (± 13.07), 4.41% (± 3.00) for processed, and 57.57% (±13.25) for ultra-processed foods (Table 2). In the distribution of the consumption of the main ultra-processed foods eaten by young people according to sleep duration, there was a difference between fast food (*p* < 0.001), salty cookies, and potato french fries (*p* < 0.000), margarine (*p* = 0.025), ketchup and mayonnaise (*p* < 0.001), and breakfast cereals and cereal bars (Appendix A).

In the crude analysis, the consumption of *in natura* or minimally processed foods (β: −0.001; 95% CI: −0.006; 0.003), processed (β: −0.009; 95% CI: −0.029; 0.010), and ultra-processed (β: 0.001; 95% CI: −0.003; 0.006) was not associated with sleep duration. In the adjusted analysis, this absence of association remained even after adjusting for confounding variables for *in natura* or minimally processed foods (β: 0000229; 95% CI: −0.004; 0.004), processed (β: −0.00282; 95% CI: −0.226; 0.169), and ultra-processed (β: 0.00003; 95% CI: −0.004; 0.005). The results showed no association between food consumption according to the degree of processing and sleep duration among adolescents (Table 3).

## 4. Discussion

This is the first study that evaluated the association between food consumption according to the NOVA classification (*in natura* or minimally processed, processed, and ultra-processed foods) [37] and sleep duration, measured by an accelerometer, in adolescents. The main results of the present study showed that the mean sleep duration was low (6 h ± 0.95) among adolescents. However, there was no association between food consumption, according to the degree of processing, and sleep duration.

The mean sleep duration of the sample of adolescents in the present study is considered low as per the National Sleep Foundation and the American Academy of Sleep Medicine, which consider sufficient sleep durations of 7 to 9 h for subjects between 18 and 64 years old [47,48]. The subjects in the present study sleep one hour less than recommended to maintain the sleep–wake cycle within the pre-established parameters. In addition, it is worth mentioning that insufficient sleep durations can lead to negative consequences for the health of adolescents, and they are associated with increased daytime sleepiness, behavioral and mental health problems, the use and abuse of alcohol and drugs, diabetes, and low immunity [49].

The results of the present study showed a mean sleep duration that is similar to the data from the study by Carone et al. [3], who analyzed 1865 university students (18 and 19 years old) in Pelotas, Rio Grande do Sul, using an accelerometer and observed a mean sleep duration of 6.5 h per night. However, this study evaluated a 24 h sleep duration on school days (during the week) and weekends, with this analysis being different for each school day and non-school day. In the present study, the assessment of sleep durations was analyzed for 24 h for up to seven consecutive days without differentiating the weekend.

On the other hand, some studies showed higher mean sleep durations. Weiss et al. [50] evaluated (through actigraphy) 240 North American subjects aged between 16 and 19 years and observed a mean sleep duration of 7.5 h. In that study, data were collected on mean sleep durations during the week and on weekends, but only data referring to five days of the week were used. Gramy et al. [1] supported these findings, determining a mean sleep duration of 7.5 h on weekdays in 278 Swedish students (15 to 17 years). However, the average assessed during weekends was higher. In the studies mentioned above, the mean sleep duration in adolescents is above the average of the present study and is consistent with the recommendations stated by international institutions for assessing sleep durations during the phases of life.

The non-association between the consumption of *in natura* or minimally processed, processed, and ultra-processed foods and sleep duration in adolescents in the present study is consistent with data from another study published in Brazil [22]. In a study of 100,648 Brazilian adolescents aged 11 to 18 years who participated in the National School Health Survey (PeNSE), Werneck et al. [22] analyzed the association between consumption of ultra-processed foods and anxiety-induced sleep disturbances. In that study, sleep duration was analyzed using a self-report questionnaire, and food consumption was assessed using a weekly food frequency questionnaire. The percentage of the consumption of ultra-processed foods in the sample was 50.0%. Although high, no association was found between the high consumption of ultra-processed foods and sleep duration. However, the consumption pattern of ultra-processed foods among adolescents is increasing; in the present study, 57.57% of the sample consumed ultra-processed foods.

Differently from the results found, Sousa et al. [4] analyzed 2499 adolescents from São Luís, Maranhão, that were aged 18 to 19 years old and found an association between food consumption according to the degree of food processing and sleep quality, evaluated by the Pittsburgh Sleep Quality Index (PSQI). In that study, most adolescents had poor sleep quality, and there was an association between in natura or minimally processed foods and a lower prevalence of poor sleep quality. In other words, the intake of in natura or minimally processed foods was a protective factor for poor sleep quality. On the other hand, consuming ultra-processed foods was associated with a higher prevalence of poor sleep quality. Therefore, the higher consumption of ultra-processed foods was a risk factor for poor sleep quality. The percentage of the intake of in natura or minimally processed foods in the sample was 57.9% for good sleep quality, while the intake of ultra-processed foods was 36.4% for poor sleep quality.

Supporting these findings, Gonçalves et al. [11] found, in a systematic review, a positive association between the intake of high-calorie ultra-processed foods and mean sleep duration. The studies selected for this research used questionnaires [51,52] and actigraphy [21,53] to assess sleep duration. Food consumption was assessed using questionnaires [21,52,53].

With respect to these observations, it is valid to point out that eating habits can impair sleep duration [10]. In this regard, many studies have shown methodological differences, and almost all of the aforementioned studies indicate that the consumption of unhealthy foods is associated with insufficient sleep durations [4,11]. However, it is essential to emphasize that a balanced diet with healthy macronutrient and micronutrient sources is essential for maintaining adequate sleep durations, especially in adolescence, a developmental stage preceding adulthood [54]. A relevant aspect to highlight is the difference between sleep duration measured by accelerometry and other sleep outcomes evaluated in different studies. Actigraphy is a suitable tool for sleep–wake detection, and it is helpful in diagnosing sleep-related disorders such as circadian rhythm disorders [55]. This device is considered a reliable sleep assessment tool by the American Academy of Sleep Medicine (AASM) [56], which is a good option compared to other outcomes that record sleep [48]. It is important to emphasize that the sleep duration assessed by accelerometry is defined, in short, as the total number of minutes without any movement at the wrist level for a given period. However, this definition is based on the absence of body movements given that the lack of body movement during the night is not the only sleep assessment attribute [31,57].

Therefore, in monitoring prolonged sleep in population studies, the accelerometer seems to be a more viable tool than polysomnography, which demands more resources and requires a laboratory environment for evaluation [31]. However, the results of sleep architecture in accelerometry do not allow for an analysis of the subjective aspects of sleep, such as sleep quality or even sleep satisfaction [57,58]. Thus, the results of the present study should be investigated with caution, given that the lack of association between *in natura* or minimally processed, processed, and ultra-processed food consumption with sleep durations may be correct for the measure used. Thus, sleep analyzed via subjective measures may be associated with food consumption.

The study has some limitations. One of these refers to collecting food consumption data using the FFQ instrument, for which its quantification can sometimes be imprecise and suffer from recall bias [51]. However, food consumption data were obtained via a valid tool for the studied population [19]. In addition, some limitations are present in the use of the NOVA classification, which can generate some difficulty in assimilating the distinction between the classifications. Therefore, in the second edition of the guide, the food groups are characterized and exemplified, including pictures, that show examples of *in natura* or minimall (fresh pineapple, corncob, and fresh fish), processed foods (pineapple in syrup, canned corn, and canned fish), and ultra-processed foods (pineapple juice powder, packaged corn snacks, and nuggets of fish); however, in some cases, this difficulty does not affect the understanding about the classification of certain foods in certain groups [59]. The other limitation may be the loss of participants who used the accelerometer due to the exclusion of data from those who did not reach the duration of sleep hours necessary for the analysis. The sample size is one of the limitations of this study. While, initially, 2515 adolescents participated in data collection, only 964 individuals had complete data regarding the use of the accelerometer. An important limitation of the study is due to the lack of an analysis on the consumption of antihistamine drugs, such as their sedative effects [60]. Another limitation is related to the sleep alterations that are noticeable in most psychiatric disorders [61]. In the International Classification of Sleep Disorders (ICSD) [62], the third division refers to sleep disorders associated with clinical and psychiatric disorders, considering that psychiatric conditions, in turn, have changes in sleep patterns as diagnostic criteria, [63] such as major depression, post-traumatic stress, and generalized anxiety disorder. Given these aspects, it is essential to point out that the study cannot have the results extrapolated relative to the population because the sample was not probabilistic.

One of the strengths is the use and construction of the DAG to identify possible confounding factors, which indicate the minimum set of variables for adjustments. This avoids unnecessary and inappropriate adjustments that do not fit the research study’s objective. In addition, it is worth noting the use of the accelerometer to monitor the sleep–wake cycle. Accelerometers are sensors that provide an electrical signal in real time, and they are very suitable for monitoring sleep duration. These devices present a desirable use in systems coupled to the body for offering objective data collection and an analysis of movement during sleep [64,65,66].

## 5. Conclusions

The findings of the study indicate that there was no association between the consumption of *in natura* or minimally processed, processed, and ultra-processed foods with sleep durations in a sample of adolescents from São Luís, Northeast Brazil. Although we did not find such associations in this study, it is important to note that the consumption of ultra-processed foods in relation to fresh and minimally processed foods is much more harmful to the health and nutrition of adolescents due to the high content of fats, refined carbohydrates, sodium, and additives. With respect to these contents, the study demonstrates that the eating habits in this age group largely comprise the consumption of ultra-processed foods. Given this fact, it is necessary to control and monitor food intake and to promote quality of life and prevent possible damage to sleep cycles and health.

## Figures and Tables

**Figure 1 nutrients-14-05180-f001:**
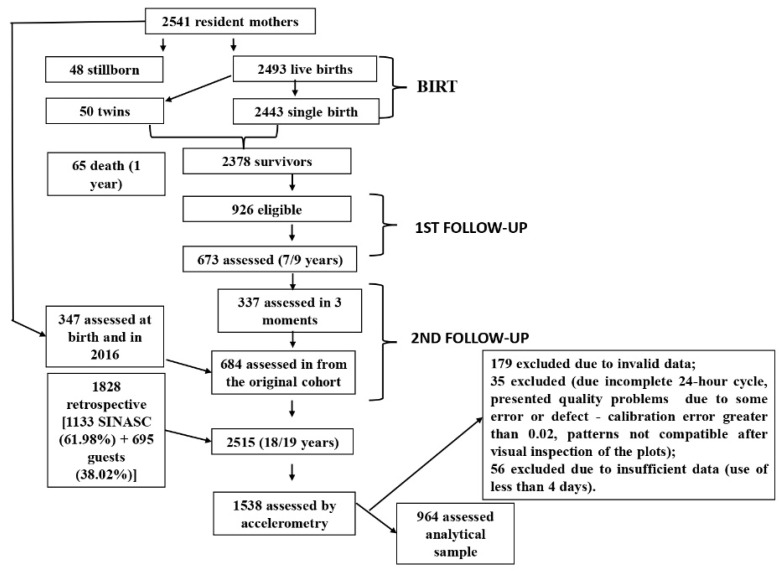
Flowchart of the 1997/1998 birth cohort of São Luís, Maranhão, Brazil.

**Figure 2 nutrients-14-05180-f002:**
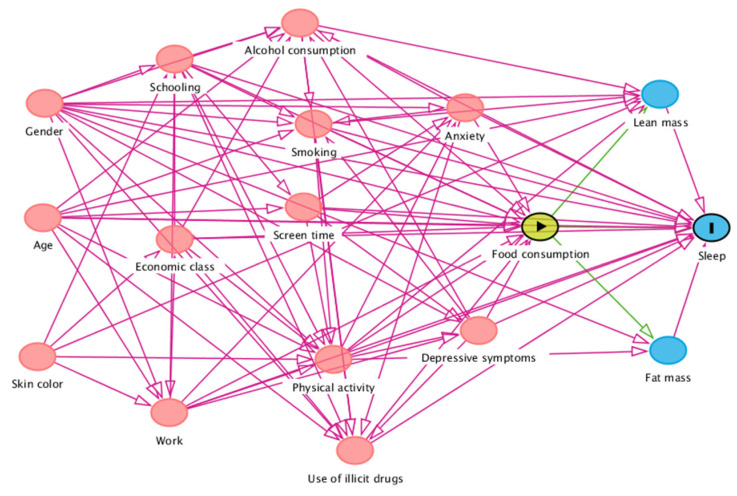
Directed acyclic graph regarding the association between food intake and sleep duration.

**Table 1 nutrients-14-05180-t001:** Sample distribution (*n* = 2515) according to the use or not of accelerometers in adolescents. São Luís, Maranhão, Brazil, 2016/2017.

Variable	Accelerometer				*p*-Value
	No		Yes		
	*n*	%	*n*	%	
Sociodemographic and economic					
Gender					0.530
Male	725	61.0	463	39.0	
Female	826	62.2	501	37.8	
Age					<0.001
18 years	914	52.5	828	47.5	
19 years	637	82.4	136	17.6	
Skin Color					0.305
White	310	62.6	185	37.4	
Black	268	64.4	148	35.6	
Brown	962	60.5	627	39.5	
Years of schooling					0.011
None to 8 years	91	66.9	45	33.1	
9 to 11 years	1238	60.4	871	39.6	
12 years or more	117	79.9	48	29.1	
Economic class					<0.001
A–B	403	61.1	257	38.9	
C	634	56.8	482	43.2	
D–E	225	50.0	225	50.0	
Family income					0.743
<1 MW	170	58.6	120	41.4	
1 MW	437	59.5	298	40.5	
2 MW	311	61.1	198	38.9	
3 MW	159	60.2	105	39.8	
≥4 MW	277	63.0	163	37.0	
Work					0.895
No	1305	61.6	813	38.4	
Yes	249	62.0	151	38.0	
Lifestyle					
Total Physical Activity					0.926
Insufficiently active	846	61.3	534	38.7	
Active	679	61.1	430	38.9	
Screen time					0.323
No use or use ≤ 2 h	137	56.0	108	44.0	
>2 to ≤5 h	324	55.3	262	44.7	
>5 h	844	58.7	594	41.3	
Consumption of alcoholic beverages					0.241
No	881	60.4	578	39.6	
Yes	649	62.7	386	37.3	
Smoking					0.256
No	1491	61.5	935	38.5	
Yes	60	67.4	29	32.6	
Use of illicit drugs					0.048
Never used	1108	60.0	737	40.0	
Have used or currently use	412	64.5	227	35.5	
Anxiety					0.598
No	1495	61.6	933	38.4	
Yes	56	64.4	31	35.6	
Depression					<0.001
No	1454	62.9	858	37.1	
Yes	97	47.8	106	52.2	
Anthropometric variables					
BMI					0.467
Adequate/Low weight	1225	61.3	773	38.7	
Overweight/obesity	326	63.1	191	36.9	
Sleep duration					
Sleep duration (in hours)					0.966
≥6 h	210	24.0	665	76.0	
<6 h	95	24.1	299	75.9	
Food consumption					
*In natura* or minimally processed	1531	61.4	964	38.6	0.485
Processed	1531	61.4	964	38.6	0.024
Ultra-processed	1531	61.4	964	38.6	0.281

Caption: MW: minimum wage; BMI: body mass index.

**Table 2 nutrients-14-05180-t002:** Characterization of the general sample and the mean sleep duration according to sociodemographic, economic, anthropometric, and behavioral characteristics in adolescents. São Luís, Maranhão, Brazil, 2016/2017.

Variable	General	Sleep Duration	
	*n* (%)	Mean (SD)	*p*-Value
Sociodemographic and economic			
Gender (*n* = 964)			<0.001 ^a^
Male	463 (48.0)	5.77 (0.93)	
Female	501 (52.0)	6.20 (0.93)	
Age (*n* = 964)			0.426 ^a^
18 years	828 (85.9)	5.99 (0.96)	
19 years	136 (14.1)	6.06 (0.92)	
Skin Color (*n* = 960)			0.776 ^c^
White	185 (19.3)	6.10 (0.94)	
Black	148 (15.4)	5.80 (0.99)	
Brown	627 (65.3)	6.00 (0.94)	
Years of schooling (*n* = 964)			0.209 ^c^
0 to 8 years	45 (4.7)	5.69 (1.06)	
9 to 11 years	871 (90.4)	6.01 (0.95)	
12 years or more	48 (4.9)	5.99 (0.82)	
Economic class (*n* = 964)			0.769 ^c^
A–B	257 (26.7)	5.97 (0.97)	
C	482 (50.0)	5.95 (0.94)	
D–E	225 (23.3)	6.13 (0.95)	
Family income (*n* = 884)			0.804 ^c^
<1 MW	120 (13.6)	6.11 (0.96)	
1 MW	298 (33.8)	5.99 (0.95)	
2 MW	198 (22.4)	6.09 (0.98)	
3 MW	105 (11.8)	5.78 (0.93)	
≥4 MW	163 (18.4)	5.99 (0.89)	
Work (*n* = 964)			0.332 ^a^
No	813 (84.3)	6.01 (0.95)	
Yes	151 (15.7)	5.93 (0.93)	
Lifestyle			
Total Physical Activity (*n* = 964)			0.007 ^a^
Insufficiently active	534 (55.4)	6.07 (0.94)	
Active	430 (44.6)	5.91 (0.96)	
Screen time (*n* = 964)			0.247 ^c^
No use or use ≤ 2 h	108 (11.2)	6.14 (0.86)	
>2 to ≤5 h	262 (27.1)	6.01 (0.93)	
>5 h	594 (61.7)	5.96 (0.97)	
Consumption of alcoholic beverages (*n* = 964)			0.435 ^a^
No	578 (59.9)	6.02 (0.93)	
Yes	386 (40.1)	5.97 (0.99)	
Smoking (*n* = 964)			0.863 ^a^
No	935 (96.9)	5.99 (0.95)	
Yes	29 (3.1)	6.03 (0.88)	
Use of illicit drugs (*n* = 964)			0.334 ^a^
Never used	737 (76.4)	6.01 (0.94)	
Have used or currently use	227 (23.6)	5.94 (1.00)	
Anxiety (*n* = 964)			0.065 ^a^
No	933 (96.8)	6.00 (0.96)	
Yes	31 (3.2)	5.92 (0.75)	
Depression (*n* = 964)			0.181 ^a^
No	858 (89.0)	5.99 (0.95)	
Yes	106 (11.00)	6.11 (0.98)	
Anthropometric variables			
BMI (*n* = 964)			0.023 ^b^
Adequate/Low weight	773 (80.2)	6.03 (0.92)	
Overweight/obesity	191 (19.8)	5.84 (1.05)	
Food consumption	Mean (SD)	Correlation	*p*-value
*In natura* or minimally processed (*n* = 964)	34.72 (13.0)	57.57 (13.25)	0.057
Processed (*n* = 964)	4.41 (3.00)	4.41 (3.00)	0.346
Ultra-processed (*n* = 964)	57.57 (13.25)	34.72 (13.07)	0.479

Caption: ^a^ Student’s *t*-test; ^b^ rank sum test; ^c^ Bartlett’s test; MW: minimum wage; BMI: body mass index.

**Table 3 nutrients-14-05180-t003:** Gross and adjusted analysis of the association of consumption of *in natura* or minimally processed, processed, and ultra-processed foods with sleep duration in adolescents. São Luís, Maranhão, Brazil, 2016/2017.

Food Consumption	Crude Analysis		Adjusted Analysis *	
	β (95%CI)	*p*-Value	β (95%CI)	*p*-Value
*In natura* and minimally processed foods	−0.001 (−0.006; 0.003)	0.570	0.000229 (−0.004; 0.004)	0.924
Processed foods	−0.009 (−0.029; 0.010)	0.346	−0.00282 (−0.226; 0.169)	0.780
Ultra-processed foods	0.001 (−0.003; 0.006)	0.479	0.00003 (−0.004; 0.005)	0.989

Caption: * Adjustment variables: gender, age, years of schooling, economic class, work, smoking, alcohol consumption, drug use, total physical activity, screen time, anxiety, and depression; β: regression coefficients; 95%CI: 95% confidence interval.

## Data Availability

The data that support the findings of this study are available from e-mail rosangela.flb@ufma.br, but restrictions apply to the availability of these data, which were used under license for the current study, and so are not publicly available. Data are however available from the authors upon reasonable request and with permission of Rosangela Fernandes Lucena Batista.

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
