# Peer review of "Association of Food Intake with Sleep Durations in Adolescents from a Capital City in Northeastern Brazil"

_nutrients, 2022, doi:10.3390/nu14235180_

Round 1
Reviewer 1 Report
This study thus aims to estimate the association of consumption of in natura or minimally processed, processed, and ultra-processed foods with sleep duration in adolescents. The rationale behind the study is clear. Unfortunately, the manuscript is not prepared according to the authors guidelines. Also, grammar and style need to be proofread by a trained medical writer to increase readability of the content.
A shortcoming of the method is that only weekday activity was measured, whereas weekend behavior in this respect would be much more interesting, also in regard to other variables such as alcohol use.
The authors stated that one of the strengths is the use and construction of the DAG, but the actual method for generating it is not sufficiently explained. The authors reported that they did a literature review on the study theme, but did not explain how.
The conclusion is not substantiated by the study findings, as there was no connection between food and sleep.
I sum, given its limited generalizability and comparability, I suggest submitting the article to more local journals. For their readers, abbreviations such as São Luís/Ma in the title are not unknow, like they are for a more international readership. This is underpinned by the fact that practical and theoretical implications are lacking, also in regard to generalizability and comparability for other countries.
Author Response
Point 1: This study thus aims to estimate the association of consumption of in natura or minimally processed, processed, and ultra-processed foods with sleep duration in adolescents. The rationale behind the study is clear. Unfortunately, the manuscript is not prepared according to the authors guidelines. Also, grammar and style need to be proofread by a trained medical writer to increase readability of the content.
Response 1: Thank you for your comment. The present study was prepared by authors from several health areas, such as medicine, nutrition, and physical education. In addition, all processes of data collection, work plan elaboration, data tabulation, methods, results analysis, discussion, conclusion, and the general writing of the manuscript were carefully reviewed by all authors. It is also worth mentioning that the team of authors includes a renowned and capable physician who is currently a full professor in the Department of Public Health and the Graduate Program in Collective Health at the Federal University of Maranhão. He has a medical degree from the Federal University of Maranhão (1984), a master's degree (1990), and a doctorate (1997) in Preventive Medicine from the Medical School of Ribeirão Preto - University of São Paulo. He did a postdoctoral fellowship in Perinatal Epidemiology at the National Perinatal Epidemiology Unit, University of Oxford, 2003. He is a CNPq 1-A researcher and editor-in-chief of the Ciência & Saúde Coletiva journal. He was a full member of the Advisory Committee of the area of Collective Health and Nutrition of CNPq (2011/2014). He was associate editor of Cadernos de Saúde Pública and of the Journal of Epidemiology and Community Health. He was a member of the Epidemiology Commission of Abrasco. He is a reviewer for several national and international journals, including: Revista de Saúde Pública, Cadernos de Saúde Pública, BMC Public Health, Paediatric and Perinatal Epidemiology, Revista Brasileira de Saúde Materno-Infantil, American Journal of Epidemiology, Social Science & Medicine, PLOS One, Pediatrics and Brazilian Journal of Medical and Biological Research. He has experience in the field of Collective Health, with emphasis in Epidemiology, working mainly on the following topics: low birth weight, infant mortality, prematurity, cesarean section and obesity. He is involved in 5 birth cohort studies, conducted in the last 25 years in São Luís and Ribeirão Preto and in the last 5 years has coordinated a cohort study of children with Congenital Zika Syndrome. He participates in 4 research consortiums: RPS (Brazilian Consortium of birth cohorts), ZBC Consortium (Brazilian Consortium of Zika cohorts), ZIKV IPD (Zika Virus Individual Patient Data Metanalysis) Consortium - World Health Organization and NCD (Non-Communicable Diseases) Risk Factor Collaboration.
Grammar and style were reviewed and adjusted by a certified translation company, and before submitting the manuscript, the author and Professor Antônio Augusto Moura da Silva reviewed it carefully to increase the readability of the content.
Point 2: A shortcoming of the method is that only weekday activity was measured, whereas weekend behavior in this respect would be much more interesting, also in regard to other variables such as alcohol use.
Point 3: The authors stated that one of the strengths is the use and construction of the DAG, but the actual method for generating it is not sufficiently explained. The authors reported that they did a literature review on the study theme, but did not explain how.
Response 3: We thank you for your comment. We have revised this part and deepened the theoretical part about DAG in the manuscript. Thus, regarding the covariates of the study, a Directed Acyclic Graph (DAG) was constructed in the DAGitty® version 3.0 program, based on the literature review on the subject of the study, which demonstrated a relationship between food intake and sleep duration (Figure 2). The DAG is called influence diagram, relevance diagram or causal network, because the arrowheads in the graph create a path between two variables. It is worth noting that among the main characteristics of these graphs are: being acyclic, that is, not allowing circularities in its composition. This means that a variable cannot interfere with itself; and to be directed, where causality follows only one direction, at a given moment in time. By assuming the temporal perspective in relation to the notion of causality, one has that time flows from left to right [45].
The DAG is a visual tool that encodes a qualitative expert knowledge or assumptions about the causal structure of a problem and serves to represent a directed causality relationship between confounding variables identified through theoretical grounding, building a minimal set for the adjustment variables [46]. Simply put, the DAG is a visual tool where it is possible to encode knowledge (from both empirical research as well as theory) regarding the causal structure of a problem. And from the observation of the created design, it is possible to identify biases (of confounding and variable selection), thus helping to define which variables should be included in the adjustment so as not to sabotage the identification/interpretation of the causal effect [47]. To investigate the association between consumption of unprocessed or minimally processed, processed and ultra-processed foods and sleep, the following minimal set of adjustment variables was selected: sex, age, skin color, education, economic class, work, alcohol consumption, smoking, screen time, physical activity, illicit drug use, anxiety, depressive symptoms, lean mass and fat mass.
[45] Greenland, S., Pearl, J., & Robins, J. M (1999) Causal diagrams for epidemiologic research. Epidemiology 37-48.
[46] Da silva, Antonio Augusto Moura. Introdução à Inferência Causal em Epidemiologia: uma abordagem gráfica e contrafactual, 1ª ed.;Editora fiocruz: Rio de Janeiro, Brasil, 2021; págs 402-402.
[47] Hernán, M. A., & Cole, S. R (2009) Invited commentary: causal diagrams and measurement bias. American journal of epidemiology 170, 959-962.
Point 4: The conclusion is not substantiated by the study findings, as there was no connection between food and sleep.
Response 4: We appreciate the comment and accept the suggestion.
Point 5: I sum, given its limited generalizability and comparability, I suggest submitting the article to more local journals. For their readers, abbreviations such as São Luís/Ma in the title are not unknow, like they are for a more international readership. This is underpinned by the fact that practical and theoretical implications are lacking, also in regard to generalizability and comparability for other countries.
Response 5: We thank you for your comment. We accept the suggestion to remove the abbreviation with São Luís/Ma from the title.
Reviewer 2 Report
Dear Editor,
I have carefully assessed the manuscript "Association of Food Intake with Sleep Duration in Adolescents 2 from the 1997/1998 Birth Cohort – São Luís/Ma". Overall, I think this study has been adequately structured and presented, and could be considered for publication after minor revisions.
- The most relevant lack of the present study is the insufficient data provided about medications assumed by the patients. This could have a major impact on the statistical analyses here provided. The authors should provide more information, on this topic, specifically for some drug categories (i.e. antihistamine medications). If this is impossible, the authors should carefully discuss it as a limitation, and the potential acceptance of this work be discussed again;
- the section in the introduction discussing the efficacy of actigraphs and accelerometers should be supported by specific literature;
- the study design is complex and a clear graph is needed in the method section to improve readability;
- data on the reliability of actigraphs in the method sections should be supported by the literature;
- the use of asterisks in the tables provided in the results to define different statistical tests could be misleading (usually here used for significance), and letters should be preferred;
- comments and classifications on the color of the skin of the included patients, if kept, should be more carefully discussed in the methods section;
- The screening tool here adopted to assess potential mental disorders provides only a slight help. Mental disorders, moreover, could potentially impact with great significance the sleep features of the included patients. This should be carefully discussed in the limitations section
Author Response
Por favor, verifique o anexo

Round 2
Reviewer 1 Report
I appreciate the modifications done in response to the first review round. However, some of them were very superficially made, e.g the abbreviation São Luís/Ma in the title was deleted, but now it is not indicated that we talk about a Brazilian study. Also, the flow chart is hardly readable due to low resolution. Although it might be true that there was no association between consumption of food with sleep duration, the long list of substantial limitations questions even this finding of a non-existing association. The use of statistical measure should be clear. E.g. it is not mentioned that table 3 is from linear regression. Also, when reporting regression results, odds ratios are more commonly used than beta.
In sum, the study shows a limited generalizability and comparability, and lacks practical and theoretical implications for international readers. E.g. although indicated in the introduction, that such information may be useful in providing health interventions for these individuals, this part was not further discussed. Thus, I suggest submitting the article to more local journals.
Round 3
Reviewer 1 Report
I thank the authors for the modifications in accordance with the previous review points. There are some issues with the formatting, e.g. figures in the text still have headings. The conclusion should include practical and theoretical implications and reflect on the significance of the findings given the limitations.
Author Response
Please see the attachment
Point 1: Please revise your manuscript according to the referees’ comments and upload the revised file within 2 days. Regarding the layout, we will do it, after you send us the revised manuscript.
Response 1: Thank you for your comment. We have adjusted the layout.
Point 2: Please use the version of your manuscript found at the above link for your revisions
Response 2: We thank you for your comment. We use the version of the manuscript found at the link for your revisions.
Point 3: Please check that all references are relevant to the contents of the manuscript
Response 3: We appreciate the comment. We have checked that all references are relevant to the content of the manuscript.
Point 4: Any revisions made to the manuscript should be marked up using the “Track Changes” function if you are using MS Word/LaTeX, such that changes can be easily viewed by the editors and reviewers
Response 4: We appreciate the comment. We make it clear that revisions made to the manuscript have been marked with MS Word's revision control, so that changes can be easily viewed by editors and reviewers.
Point 5: Please provide a short cover letter detailing your changes for the editors’ and referees’ approval.
Response 5: We appreciate the comment. We will make a short cover letter detailing your changes for the editors' and referees' approval.
Point 6: There are some issues with the formatting, e.g. figures in the text still have headings.
Response 6: We thank you for your comment. We have adjusted the layout of the manuscript and adjusted the figure titles.
Point 7: The conclusion should include practical and theoretical implications and reflect on the significance of the findings given the limitations.
Response 7: Thank you for your comment. We accept your suggestion and in the conclusion we will write as follows: “The findings of the study indicate that there was no association between the consumption of unprocessed or minimally processed, processed and ultra-processed foods with sleep duration in a sample of adolescents from São Luís, Northeast Brazil. Although we did not find such associations in this study, it is important to note that the consumption of ultra-processed foods in relation to fresh and minimally processed foods is much more harmful to the health and nutrition of adolescents, due to the high content of fats, refined carbohydrates, sodium and additives. About this, the study demonstrates that the eating habits in this age group largely comprise the consumption of ultra-processed foods. Given this fact, it is necessary to control and monitor food intake and to promote quality of life and prevent possible damage to the sleep cycle and health.”
